# Fast Frequency-Diverse Radar Imaging Based on Adaptive Sampling Iterative Soft-Thresholding Deep Unfolding Network

Zhenhua Wu [1,2,3], Fafa Zhao [1], Lei Zhang [4,*], Yice Cao [1], Jun Qian [1], Jiafei Xu [1] and Lixia Yang [1]

1 Information Materials and Intelligent Sensing Laboratory of Anhui Province, Anhui University, Hefei 230601, China; zhwu@ahu.edu.cn (Z.W.); p20301162@stu.ahu.edu.cn (F.Z.); yccao@ahu.edu.cn (Y.C.); p21201071@stu.ahu.edu.cn (J.Q.); p21301160@stu.ahu.edu.cn (J.X.); 19002@ahu.edu.cn (L.Y.)
2 State Key Laboratory of Complex Electromagnetic Environment Effects on Electronics and Information System, Luoyang 471000, China
3 State Key Laboratory of Millimeter Waves, Southeast University, Nanjing 210096, China
4 School of Electronics and Communication, Sun Yat-Sen University, Guangzhou 510275, China
* Correspondence: zhanglei57@mail.sysu.edu.cn

**Abstract:** Frequency-diverse radar imaging is an emerging field that combines computational imaging with frequency-diverse techniques to interrogate the high-quality images of objects. Despite the success of deep reconstruction networks in improving scene image reconstruction from noisy or under-sampled frequency-diverse measurements, their reliance on large amounts of high-quality training data and the inherent uninterpretable features pose significant challenges in the design and optimization of imaging networks, particularly in the face of dynamic variations in radar operating frequency bands. Here, aiming at reducing the latency and processing burden involved in scene image reconstruction, we propose an adaptive sampling iterative soft-thresholding deep unfolding network (ASISTA-Net). Specifically, we embed an adaptively sampling module into the iterative soft-thresholding (ISTA) unfolding network, which contains multiple measurement matrices with different compressed sampling ratios. The outputs of the convolutional layers are then passed through a series of ISTA layers that perform a sparse coding step followed by a thresholding step. The proposed method requires no need for heavy matrix operations and massive amount of training scene targets and measurements datasets. Unlike recent work using matrix-inversion-based and data-driven deep reconstruction networks, our generic approach is directly adapted to multi-compressed sampling ratios and multi-scene target image reconstruction, and no restrictions on the types of imageable scenes are imposed. Multiple measurement matrices with different scene compressed sampling ratios are trained in parallel, which enables the frequency-diverse radar to select operation frequency bands flexibly. In general, the application of the proposed approach paves the way for the widespread deployment of computational microwave and millimeter wave frequency-diverse radar imagers to achieve real-time imaging. Extensive imaging simulations demonstrate the effectiveness of our proposed method.

**Keywords:** adaptive sampling; deep unfolding; data driven; ASISTA-Net; model driven; frequency diverse; radar imaging

## 1. Introduction

With the ability to penetrate most optically opaque materials and achieve non-ionizing radiation, microwave imaging [1,2] has many advantages over other modalities of imaging, such as CT [3], X-ray [4], MRI [5], particularly in security screening scenario [6]. Despite the superiority, there is still a challenge, that is, the need to synthesize composite apertures to scan the scene and acquire high resolution either mechanically or electronically. Specifically, scene information acquisition with mechanical scanning is relatively time consuming and not suitable for real-time operation, while electronic scanning with phased arrays can be complex and power intensive. To overcome this challenge, researchers have explored

various approaches that leverage advances in computational methods and component technologies. One promising approach is the compressed sensing (CS) technique [7,8], which uses optimization algorithms to reconstruct an image from the under-sampled measurements that would normally be required, thus yielding a significantly reduced data acquisition time and achieving near real-time operation. In terms of antenna hardware platform, the compact and low-profile metasurface antennas without complex phase-shifting circuits or power amplifiers are capable of manipulating the electromagnetic waves and achieving beam steering and have received considerable research attention.

Typically, frequency-diverse radar imaging is a highly effective computational imaging technique that employs frequency-diverse antennas to capture and reconstruct scene information. The radiation fields of a frequency-diverse antenna exhibit quasi-random or quasi-orthogonal variation over a given frequency bandwidth. Frequency-diverse imaging attains an advantage by encoding scene information onto quasi-random field patterns, which are obtained by stepping through several frequency points within the radar operating frequency band. These field patterns are unique to each frequency, and by capturing the full set, the complete information of the scene can be reconstructed. The effectiveness of this technique is well documented in several works [1,9–12]. This means that data acquisition can be performed in an all-electronic manner, without the need for mechanical scanning or complex phase-shifting components. To reconstruct an image of the scene from the acquired data, computational techniques are used to interact the measurements with the transfer function of the frequency-diverse imaging system. Direct algorithms, such as the matched-filter technique [13], least-square technique [14,15], and other sparsity-driven optimization methods [16] are commonly used to reconstruct a high-quality image of the scene.

In recent years, the deep neural reconstruction network has rapidly been emerging in radar imaging literature as an extremely powerful technique for solving high-complexity reconstruction and imaging problems with unprecedented computational efficiency without sacrificing accuracy and reliability [17–20]. With the abilities to learn and extract features from large datasets in combination with the mighty non-linear fitting capabilities, deep reconstruction networks have been widely used in image restoration [21], super-resolution imaging [22], image denoising [23], and medical imaging fields [24]. Specifically, there are mainly two representative types of deep reconstruction network:

(1) Data-driven deep reconstruction network: By training on large datasets of high-quality scene targets and measurements, the underlying non-linear relationship between the acquired measurements and the reconstructed scene targets can be directly learned by the deep neural network. The model-driven algorithm employs the network architecture's feedforward capabilities to format images, removing iterations from the imaging process. Adaptive network parameter adjustments take place through the use of training data. The trained networks can thus be used to obtain scene targets given the echo signal, among which, particularly, the fully convolutional neural networks (FCNs) [25] and UNet [26] and deep residual networks [27] have been well utilized for image formation in sparse SAR and ISAR imaging [28–30].

(2) Model-driven approach: Aiming at avoiding iterations optimization and sophisticated regularization parameters turning, model-driven methods [31–33] are built based on deep unfolding techniques that stem from the standard linear optimization algorithms, including IHT/IST [31] and ADMM networks [32] and AMP networks [33]. Each iteration of the algorithm is represented as a layer in the neural network, creating a deep network that performs a finite number of algorithm iterations when passing through. During backpropagation training, a number of model parameters of the algorithm can be converted to network parameters, resulting in a highly parameter-efficient network. In general, model-driven methods provide a promising direction for interpreting and optimizing iterative algorithms in combination with deep neural networks [34,35]. Overall, data-driven deep reconstruction neural networks are highly dependent on the abundance and multiplicity of training data, while, in comparison,

data-driven methods with the unfolding technique could effectively use the training data and still maintain preferable image formation performance with limited amounts of training data.

With the collected training echo data and scene images, data-driven reconstruction neural networks aim at learning the optimal mapping relations corresponding to the measurement matrix (radiation or illumination patterns), which is highly task specific and requires retraining when the radar working frequency band changes.

Data-driven reconstruction neural networks leverage collected training echo data and scene images to learn optimal mapping relations specific to the measurement matrix (radiation of illumination patterns) [10]. However, these networks often require retraining when the radar working frequency band is changed, as the mapping relations are highly task specific. To address this limitation, in this paper, we aim to reduce the computational complexity of real-time imaging while achieving flexible radar working frequency usage. Drawing inspiration from the recent advancements in *beyond deep unfolding* reconstruction techniques [34–37], we introduce the adaptive sampling iterative soft-thresholding deep unfolding network (ASISTA-Net). By leveraging the ISTA unfolding layer as the backbone reconstruction network, we replace each iterative process with end-to-end convolution layers. Additionally, we incorporate an adaptive sampling module into the reconstruction network. Our approach offers the ability to train measurement matrices at different sampling ratios in parallel, enabling frequency-diverse radar to utilize various working frequency bands through a single training. Unlike traditional linear optimization methods, our proposed method effectively handles high correlation in sensing matrices, even at extremely low scene sampling ratios. Notably, our approach eliminates the need for sequential training when the sensing matrix is changed. To validate the effectiveness and robustness of our method, we conduct extensive imaging simulations using synthesis radiation field data. The results demonstrate the superior performance of ASISTA-Net in handling real-time imaging with reduced computational complexity, flexible frequency utilization, and adaptability to different scene-sampling ratios.

## 2. Imaging Principle

By developing a series of subwavelength resonant units with simple structures, the integrated antenna with frequency diversity can successfully manipulate the polarization properties of the electric and magnetic dipoles, thereby enabling the precise measurement and analysis of the target. In this near-field computational imaging framework, the measurement matrix is established by calculating the product of electric fields at each position in the scene from both the transmitting antenna and the receiving probe, and then correlating the resulting frequency measurements with the target scattering coefficients.

In principle, frequency-diverse computational imaging is actually a process of the traditional inverse scattering problem. The receiving antenna collects frequency measurements which are related to the scattering coefficient of the target area by a transfer function. This transfer function is determined by the transmission and receiving properties of the antennas at each position in the target area. The ratio of the total field in the open waveguide is determined by this transfer function:

$$\mathbf{g}(f) = \int_T \mathbf{U}_{\text{oTX}}(\vec{\mathbf{r}}'; f) \mathbf{U}_{\text{oRX}}(\vec{\mathbf{r}}'; f) \sigma(\vec{\mathbf{r}}') \mathrm{d}^2 \vec{\mathbf{r}}' \tag{1}$$

where the transmit and receive fields at frequency $f$ generated by the transmit and receive apertures at scene location $\vec{r}$ are represented by $\mathbf{U}_{\text{oTX}}(\vec{\mathbf{r}}'; f)$ and $\mathbf{U}_{\text{oRX}}(\vec{r}'; f)$, respectively. The scene target scattering coefficient is represented by $\sigma$. Given the imaging system's constrained bandwidth, we express Equation (1) as a more general and compact matrix equation:

$$\mathbf{g} = \mathbf{H}\sigma + \mathbf{n} \tag{2}$$

Equation (2) describes the imaging measurement process for a frequency-diverse antenna. Here, $\mathbf{g} \in \mathbb{C}^{M \times 1}$ represents the received echo vector of the probe antenna, with $M$ denoting the number of effective radiation direction maps. The scattering coefficient of the scene to be resolved is represented by $\sigma \in \mathbb{C}^{N \times 1}$, where $N$ is the number of discrete units in the scene. The measurement noise term is included as $\mathbf{n} \in \mathbb{C}^{M \times 1}$. The measurement matrix or measurement mode is represented by $\mathbf{H} \in \mathbb{C}^{M \times N}$ and describes the relationship between the transmitting and receiving antenna fields. In practice, a first-order Born approximation model is typically used to accurately represent the propagation process of the incident and scattered fields. In this paper, we propose new imaging methods that leverage this imaging measurement equation to improve upon existing techniques.

In Equation (2), once the discomfort problem has been solved by recovering the original scene target scattering coefficient, $\sigma$, from the collected echo $\mathbf{g}$, a sparse representation of the echo signal can be employed to solve the reconstruction problem using the parametric minimization method:

$$\min_{\sigma} \|\sigma\|_{l_1} \quad s.t. \quad \mathbf{g} = \mathbf{H}\sigma \tag{3}$$

Further, the above Equation (3) can be transformed into an L1 regularized minimization model based on a convex relaxation algorithm:

$$\hat{\sigma} = \arg\min_{\sigma} \frac{1}{2}\|\mathbf{g} - \mathbf{H}\sigma\|_2^2 + \lambda\|\mathbf{\Psi}\sigma\|_1 \tag{4}$$

where $\sigma$ denotes the frequency-diverse target scene to be solved, $\lambda$ denotes the regularized coefficients, $\mathbf{\Psi}$ denotes the sparse transformation of $\sigma$, and the regularization containing the $l_1$ parametrization constrains $\mathbf{\Psi}_\sigma$ to some extent, converting the original pathological inverse problem into an approximate fitness problem to be solved.

Using $\phi(\sigma)$ to represent a generic regularization term, the above equation can be described as

$$\hat{\sigma} = \arg\min_{\sigma} \frac{1}{2}\|\mathbf{H}\sigma - \mathbf{g}\|_2^2 + \phi(\sigma) \tag{5}$$

The traditional iterative soft thresholding algorithm ISTA generally solves the above reconstruction problem iteratively by the following steps:

$$\mathbf{r}^k = \sigma^{(k-1)} - \rho\mathbf{H}^{\mathrm{T}}(\mathbf{H}\sigma^{(k-1)} - \mathbf{y}) \tag{6}$$

$$\sigma^k = \arg\min_{\sigma} \frac{1}{2}\left\|\sigma - \mathbf{r}^k\right\|_2^2 + \lambda\|\mathbf{\Psi}\sigma\|_1 \tag{7}$$

where $k$ denotes the iteration index value and $\rho$ denotes the iteration step length. In the whole iterative solution process, the sparse transform base phi and the parameters $\lambda$ and $\rho$ are set manually in advance, and are continuously updated and adjusted according to the reconstruction results during the iterative solution process, which leads to the disadvantages of the traditional ISTA algorithm, such as high computational complexity and redundancy of iterative update steps.

## 3. Imaging Network Model

### 3.1. Imaging Network Framework

Deep imaging networks that currently exist are generally composed of several layers and require large amounts of data to be able to train the model effectively. These networks are constructed without incorporating algorithm-driven physical models, and thus, possess disadvantages, such as poor generalization ability, slow network convergence, and dependence on a large amount of data for network training. Thus, we propose a parametric adaptive sampling frequency fractionated imaging network (ASISTA-Net) in this paper. Based on a model-driven depth unfolding network, each reconstructed fraction in the imaging model is mapped by employing the specific iterative process of the traditional soft threshold iterative method. To significantly enhance the network's reconfiguration

performance, expand its capacity and its data-processing capabilities, the deep network architecture incorporates a sparse transformation step. The sparse transformation of the radar echo data in ASISTA-Net is performed by the use of the sparse transform base $\mathbf{\Psi}$ to yield a sparse optimal solution for frequency-diverse antenna imaging. We denote the sparse transform process as $\mathcal{T}(\cdot)$, and it replaces the original manually defined transform base $\mathbf{\Psi}$.

After replacing the sparse basis $\mathbf{\Psi}$ with $\mathcal{T}(\cdot)$, a new linearly transformed sparse regularization problem can be obtained as follows:

$$\hat{\sigma} = \arg\min_{\sigma} \frac{1}{2}\|\mathbf{H}\sigma - \mathbf{g}\|_2^2 + \lambda\|\mathcal{T}(\sigma)\|_1 \tag{8}$$

Thus, the iterative solution step of the above equation can be obtained by ISTA:

$$\mathbf{r}^k = \sigma^{(k-1)} - \rho\mathbf{H}^{\mathrm{T}}(\mathbf{H}\sigma^{(k-1)} - \mathbf{g}) \tag{9}$$

$$\sigma^k = \arg\min_{\sigma} \frac{1}{2}\left\|\sigma - \mathbf{r}^k\right\|_2^2 + \lambda\|\mathcal{T}(\sigma)\|_1 \tag{10}$$

where (8) represents a mathematical model that transforms the solution method of the CS-based convex relaxation algorithm into an $l_1$ regularized minimization model, which is used to obtain the imaging result $\hat{\sigma}$. This imaging model replaces the linear transformation $\mathbf{\Psi}$ with the nonlinear transformation $\mathcal{T}$ using the convolution operation, and the proposed network incorporates the concepts and principles from (8), using them as a foundation to design the network's structure, connectivity, and training methodology. The network aims to effectively capture the essence of (8)–(10) and apply it in a practical, trainable framework.

*3.2. Imaging Reconstructed Algorithm*

The ability of CNN to extract and characterize data features allows us to design $\mathcal{T}(\sigma)$ by combining two convolutional layers and using an activation layer as a separator between them. The convolutional layer conducts a local correlation scanning process on the forward input data, which completes feature extraction and removes redundant information. An activation function called Leaky_ReLU is used in the activation layer. This function corrects the results of the convolutional layer by preventing the network from overfitting, while simultaneously mapping information to the next convolutional layer for further processing.

Figure 1 demonstrates our proposed ASISTA-Net imaging network framework. The proposed imaging model is composed of two modules: the adaptive sampling module and the reconstruction imaging module. The adaptive sampling module first obtains the target scattering coefficient $\sigma$ after column-wise quantization of the scene target. It obtains the echo measurement value by the forward imaging model from the obtained measurement matrix corresponding to different sampling compression ratios and then initializes to obtain the estimated value of the target scattering coefficient $\sigma^0$. The reconstruction module divides the obtained $\sigma^0$ into real and imaginary parts for optimization training, respectively. After N convolution optimization modules, the reconstructed real and imaginary parts of the target are obtained, and a reconstructed assigned target is obtained through the adhesion function, which is the imaging result of the network.

To achieve the specific iterative update process for the mapping of the convolutional layers, the specific solution of $\mathbf{r}^k$ and $\sigma^k$ in (6) and (7) is chosen to be implemented in two parts, which constitute each layer of the imaging network reconstruction module. This module contains two convolutional layers: a Leaky_ReLU activation layer and a BN layer. To enhance the effectiveness of the imaging network, the number of convolutional kernel channels in the convolutional layer is set to 32, and the convolutional kernel size is $3 \times 3$. The batch_size is 16 as expressed in detail in Figure 1 for this part.

In the L-th layer of the imaging network, the $\mathbf{r}^k$ module corresponding to equation ($\mathbf{r}^k$) is the corresponding preliminary reconstruction result $\mathbf{r}^k$, and the $\mathbf{H}^{\mathrm{T}}(\mathbf{H}\sigma^{(k-1)} - \mathbf{g})$ is obtained by calculating the gradient of $\frac{1}{2}\|\mathbf{H}\sigma - \mathbf{g}\|_2^2$. In addition, the step size $\rho$ changes with the continuous iterative process of the network. This operation can increase the flexibility and versatility of the network. When the input is $\sigma^{k-1}$, the $\mathbf{r}^k$ input of the module can be obtained:

$$\mathbf{r}^k = \sigma^{(k-1)} - \rho^k \mathbf{H}^{\mathrm{T}}(\mathbf{H}\sigma^{(k-1)} - \mathbf{g}) \tag{11}$$

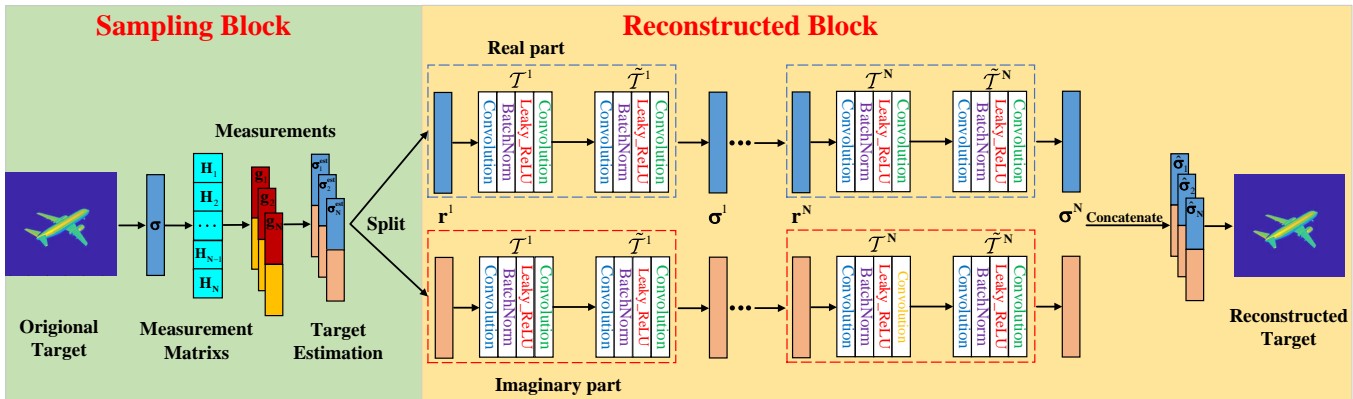

**Figure 1.** ASISTA-Net system structure diagram.

Typically, in the process of solving computational imaging problems, it can be assumed that each element of $\sigma^k - \mathbf{r}^k$ obeys an independent normal distribution and has a common mean and variance $\varepsilon^2$. Further assuming that $\mathcal{T}(\mathbf{r}^k)$ and $\mathbf{r}^k$ are the means of $\mathcal{T}(\sigma^k)$ and $\sigma^k$, respectively, the following relationship can be obtained:

$$\left\|\mathcal{T}(\sigma^k) - \mathcal{T}(\mathbf{r}^k)\right\|_2^2 \approx \varepsilon \left\|\sigma - \mathbf{r}^k\right\|_2^2 \tag{12}$$

Substituting the above equation into the $\sigma^k$ solving process of Equation (10) yields

$$\sigma^k = \arg\min_\sigma \frac{1}{2}\left\|\mathcal{T}(\sigma^k) - \mathcal{T}(\mathbf{r}^k)\right\|_2^2 + \gamma \left\|\mathcal{T}(\sigma^k)\right\|_1 \tag{13}$$

where $\lambda$ and $\varepsilon$ can be combined into a single parameter $\gamma = \lambda\varepsilon$ so that the closed form of $\mathcal{T}(\mathbf{r}^k)$ can be obtained as

$$\mathcal{T}(\sigma^k) = soft(\mathcal{T}(\mathbf{r}^k), \gamma) \tag{14}$$

Second, since the invertibility property of the sparse transform leads to a closed solution for $\sigma^k$, $\widetilde{\mathcal{T}}(\cdot)$ is introduced to act as the left inverse of $\mathcal{T}(\cdot)$ while always satisfying that the two can be combined into a constant operator denoted as $\widetilde{\mathcal{T}} \circ \mathcal{T} = \mathcal{I}$. In the network framework, since $\widetilde{\mathcal{T}}(\cdot)$ is the left inverse of $\mathcal{T}(\cdot)$, it is designed as a symmetric structure with respect to $\mathcal{T}(\cdot)$, then $\widetilde{\mathcal{T}}(\cdot)$ and $\mathcal{T}(\cdot)$ are modeled as two symmetric linear operators, and the linear operator can be efficiently derived by the following closed form:

$$\sigma^k = \widetilde{\mathcal{T}}(soft(\mathcal{T}(\mathbf{r}^k), \gamma)) \tag{15}$$

In the network training, $\gamma$ as the shrinkage threshold is a learnable parameter in this module; accordingly, in order to extend the network capacity and increase the network processing power, it is not required that $\widetilde{\mathcal{T}}(\cdot)$, $\mathcal{T}(\cdot)$ and $\gamma$ are the same for each layer, that is, each layer of the network has its corresponding $\mathcal{T}(\cdot)$, $\widetilde{\mathcal{T}}$, $\gamma$. Therefore, at the L-th layer of the network, for all learnable parameters, the output $\sigma^k$ of the module should be

$$\sigma^k = \widetilde{\mathcal{T}}^k(soft(\mathcal{T}(\mathbf{r}^k), \gamma^k)) \tag{16}$$

In Figure 1, ASISTA-Net consists of an N-layer network, and the forward transform $\mathcal{T}(\cdot)$ is designed as a combination of two convolutional layers separated by an activation layer and a BN layer, while the structure of the backward transform $\widetilde{\mathcal{T}}$ is symmetric with it and always satisfies $\widetilde{\mathcal{T}} \circ \mathcal{T} = \mathcal{I}$.

The learnable parameters set in the ASISTA-Net imaging network are $\theta$, which include the step size $\rho^k$ in the $\mathbf{r}^k$ module, the parameters of the $\mathcal{T}(\cdot)$ and $\widetilde{\mathcal{T}}$ forward and backward transforms, and the shrinkage threshold $\gamma^k$ in the $\sigma^k$ module, and thus $\theta = \rho^k$, where N is the total number of network layers in ASISTA-Net, and these parameters are learned during the network training. In addition, both $\mathcal{T}(\cdot)$ and $\widetilde{\mathcal{T}}$ are also learnable throughout the network training process, and the sparse transformation process is automatically defined by these two learnable operators. Algorithm 1 shows the specific training update process of ASISTA-Net network.

---

**Algorithm 1** ASISTA-Net training algorithm.

---

**Input:** **H**: measurement matrix; T: maximum training epochs; $\{\mathbf{g}_1, \mathbf{g}_2, \ldots \mathbf{g}_m\}$ : Echo signal testing dataset; $\{\sigma_1, \sigma_2, \ldots \sigma_n\}$ : Scene target training dataset; $\mathbf{\Psi}$: Sparse transform basis; $\rho$: step size ; $\lambda$: optimization parameter.

1: Initialize the network weights parameter $\theta$;

2: Iteration via a gradient descent scheme:

3: **for** T **do**

4: 　　Sampling a batch of s training samples $\{\sigma_1, \sigma_2, \ldots \sigma_s\}$

5: 　　For the i-th training sample, calculate $\mathbf{r}^k = \sigma^{(k-1)} - \rho \mathbf{H}^{\mathrm{T}}(\mathbf{H}\sigma^{(k-1)} - \mathbf{g})$; $\sigma^k = \arg\min_\sigma \frac{1}{2}\left\|\sigma - \mathbf{r}^k\right\|_2^2 + \lambda\|\mathcal{T}(\sigma)\|_1$

6: 　　The gradient is optimally updated by Adam's algorithm and the loss function is calculated as follows: $\frac{1}{n}\sum_{i=1}^n\|\tilde{\sigma}_i - \sigma_i\|$.

7: **end for**

**Output:** The target reflection coefficient estimate $\hat{\sigma}$.

---

## 4. Numerical Tests

### 4.1. Data Pre-Processing

The experimental dataset is preprocessed using MATLAB 2022a. The imaging technique builds upon the TensorFlow2.6 deep learning network framework, with the Keras deep learning framework and TensorFlow backend platform used to construct the model within a Python programming environment. Training involves the utilization of a machine equipped with an NVIDIA 4070 GPU, with CUDA version 11.8 installed. The ADAM gradient optimizer adjusts the entire image network to a learning rate of $1 \times 10^{-4}$. The imaging network has a 7-layer convolutional module with a $3 \times 3$ kernel size, and a step size of 1. The leaky_relu function activates each convolutional layer, allowing back-propagation for negative input, which re-balances the complexity and enhances the imaging network's reconfiguration performance. All image reconstruction results are run on a computer with an Intel(R) Core(TM) i7-12700 CPU, with batch preprocessing of the dataset performed on the MATLAB platform. Without a GPU, it takes 15 h to train the model for 200 epochs with the given dataset. However, it takes nearly 2.5 h on a GPU.

### 4.2. Imaging Parameters

In this section, we verify the effectiveness of the proposed ASISTA-Net imaging network by utilizing an uneven dataset. The training dataset consists of 1000 samples, each sample having a dimension of $33 \times 33$. The testing set aims to evaluate the matured imaging network's performance on targets such as letters, planes, clothes, and other complex scene targets.

Moreover, the ASISTA-Net's effectiveness was established by measuring the radiated field data. To demonstrate its near-field CI capability, we constructed a two-dimensional parallel-plate waveguide super-surface antenna with a waveguide slit-feed mechanism. To

obtain backscattered signals from all directions and frequencies, we employed an open-end waveguide (OEWG) probe as a receiving antenna. The antenna's panel size was $250 \times 250$ mm$^2$. The waveguide's panel-to-probe configuration had a dielectric constant of 3.66, a loss tangent of 0.003, and a substrate thickness of 0.5 mm between the copper ground plane and the conductive copper metamaterial hole. The top conductor of the waveguide had $125 \times 125$ cELC metamaterial resonators, each with a Q factor ranging from 50–60. Table 1 lists the antenna's system parameters. We performed imaging experiments on the simulated directional maps and imaging scenarios of the superlattice surface antenna radiation field, operating within a 5 MHz frequency sampling interval of 33–37 GHz. The directional maps were sampled along a two-dimensional spherical coordinate system of elevation and azimuth, with a field-of-view (FOV) size of $(-60°–60°)$ in elevation and azimuth, respectively, using a sampling interval of 2°. The resulting sampling intervals produced the original pattern T of size $800 \times (61 \times 61)$.

**Table 1.** Main system parameters of frequency-diverse antenna.

| Parameters | Values |
| --- | --- |
| Operation bandwidth | 33–37 GHz |
| Antenna panel size | $250 \times 250$ mm$^2$ |
| Number of resonance units | $125 \times 125$ |
| Frequency sampling interval | 5 MHz |
| Field of view (Azimuth) | $-60°–60°$ |
| Field of view (Elevation) | $-60°–60°$ |
| Azimuth sampling interval | 2° |
| Elevation sampling interval | 2° |
| Dimensions of **T** | $800 \times 3721$ |

Additionally, we utilized the mean squared error (MSE), peak signal-to-noise ratio (PSNR), and structural similarity index (SSIM) to evaluate the imaging performance of the network. The formulas for these metrics are detailed below:

$$\text{MSE} = \sum_{i=1}^{m} \frac{(\sigma_i - \hat{\sigma}_i)^2}{m} \tag{17}$$

$$\text{PSNR} = 10 \log_{10}\left(\frac{MAX_I^2}{\text{MSE}}\right) \tag{18}$$

$$\text{SSIM} = \frac{(2\mu_{\sigma_i}\mu_{\hat{\sigma}_i} + C_1)(2\delta_{\sigma_i\hat{\sigma}_i} + C_2)}{(\mu_{\sigma_i}^2 + \mu_{\hat{\sigma}_i}^2 + C_1)(\delta_{\sigma_i}^2 + \delta_{\hat{\sigma}_i}^2 + C_2)} \tag{19}$$

where it is assumed that the maximum possible pixel value of the images is 1 since the pixel values $MAX_I^2$ of the target image are normalized to the range of 0–1. Moreover, MSE, SSIM and PSNR formulas involve the means ($\mu_{\sigma_i}$ and $\mu_{\hat{\sigma}_i}$), variances ($\delta_{\sigma_i}$ and $\delta_{\hat{\sigma}_i}$), and covariance ($\delta_{\sigma_i\hat{\sigma}_i}$) between the original target and reconstructed target, as well as two constants ($C_1$ and $C_2$) to prevent division by zero.

### 4.3. Numerical Tests

In order to verify the effectiveness of the proposed ASISTA-Net imaging network, we carefully selected several different scene targets for imaging simulation experiments, and the imaging results are shown in Figure 2. What can be seen is that the proposed ASISTA-Net can reconstruct the original targets well in several scene targets, whether they are simple sparse targets or surface targets with complex structures, and the reconstructed feature information is very obvious.

To investigate the reconstruction performance under various compression ratios, a set of "airplane" test targets is chosen to reconstruct images using different imaging methods. The ADAM algorithm is selected to train the gradients, with a learning rate of $1 \times 10^{-4}$

to enable better control over the weight update rate and achieve superior performance when the training converges. The number of learning cycles is 200 epochs, as an increase in the number of training sessions promotes more thorough network training, leading to better imaging performance. The proposed algorithm is further evaluated by testing with compression ratios ranging from 0.05, 0.1, 0.15, and 0.2. Comparison experiments are performed using various methods, including the traditional ISTA method, sparse Bayesian learning (SBL) algorithm, VAE [38], CCU-Net methods [26], and the proposed ASISTA-Net, all under the same conditions as the proposed algorithm to identify targets. The imaging simulation results are presented in Figure 3.

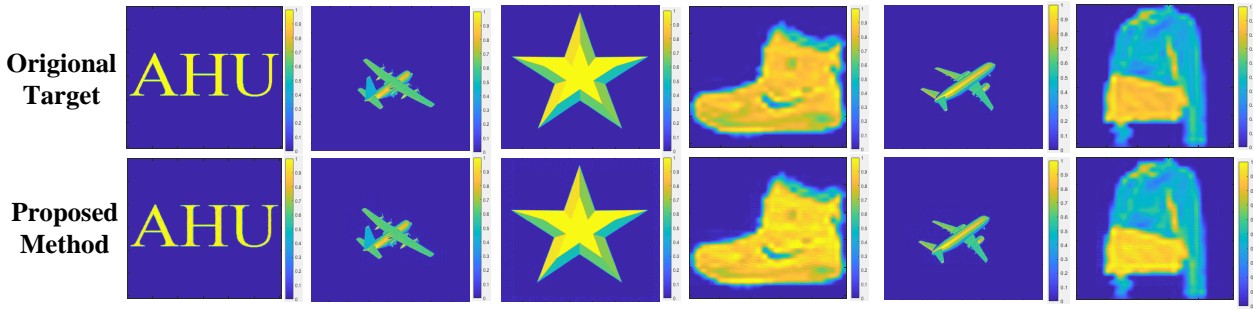

**Figure 2.** Reconstructed results from ASISTA-Net with different scene target.

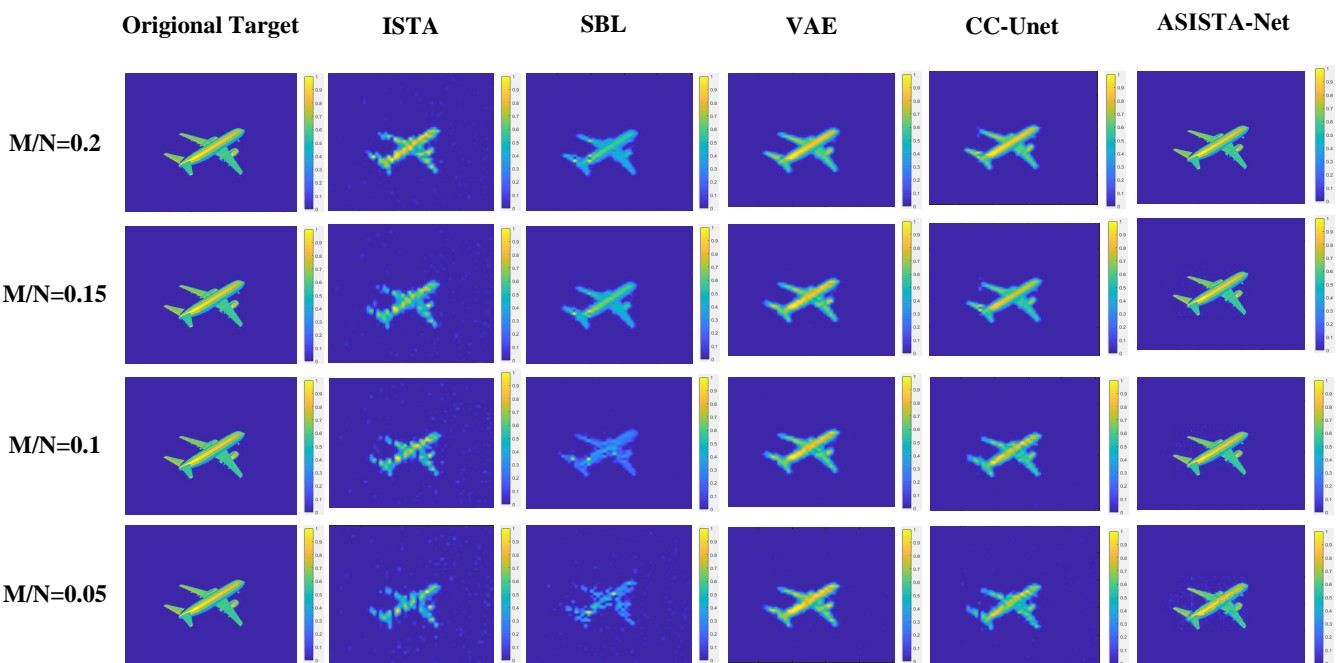

**Figure 3.** Reconstructed results in five imaging algorithms with different scene information sampling ratio.

The experimental results demonstrate that the proposed ASISTA-Net effectively reconstructs the target image at a scene information sampling rate of 0.1. Moreover, even as the scene information sampling compression ratio decreases, our algorithm still produces satisfactory reconstructions of the basic shape of the scene target, even at a much lower sampling rate of 0.05. On the other hand, at the scene information sampling rate of 0.1, the ISTA and SBL algorithms perform inadequately, with the reconstructed target image being unacceptable. The overall contour shape of the target is not completely reconstructed, and the reconstructed image exhibits poor quality and accuracy. These observations show the limitations of ISTA and SBL algorithms when working with very low scene information

compression ratios, with the imaging results lacking details of the original target. In Figure 3, while the reconstructed imaging results of VAE and CCU-Net approximate the shapes of the targets, they are unable to capture specific features of the target in detail. Overall, when compared against conventional algorithms, the proposed algorithm performs more robustly and efficiently, particularly for low scene compression ratios and high coherence measurement modes. Consequently, the reconstructed target images demonstrate high quality and can withstand challenges encountered in complex scene conditions.

Quantitatively, the MSE and PSNR values of the different methods under different scene information compression ratios were recorded simultaneously during the above imaging process, and these results are plotted in Figure 4a,b. As expected, the conventional ISTA and SBL methods performed poorly when compared to other methods in terms of the two sets of metric values, regardless of the scene information compression ratio, and the results were worse. For VAE and CCU-Net, the performance of these two methods is better under each set of compression ratios but still lower than that of our proposed ASISTA-Net. In general, as the compression ratio changes, the proposed ASISTA-Net imaging algorithm shows greater imaging capability than several other methods.

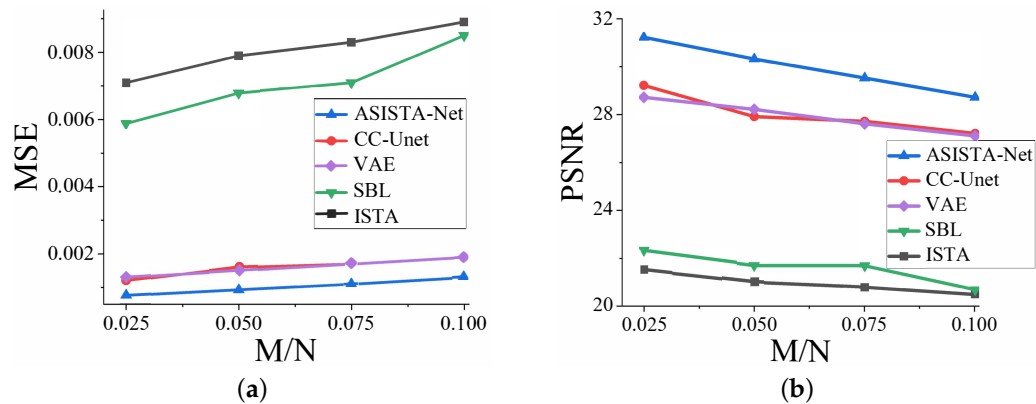

**Figure 4.** Imaging results with different scene information sampling ratio. (**a**) MSE performance. (**b**) PSNR performance.

Ten imaging tests using different imaging methods were performed on the same set of test targets. The average MSE, PSNR, SSIM, and imaging runtime for each imaging method are presented in Table 2. Additionally, to exploit the parallelizability of neural network-based methods, the reconstruction time was monitored while performing the proposed method on a GPU. According to Table 2, the proposed algorithm of ASISTA-Net outperforms several other methods. This is mainly because the end-to-end model-driven unfolding network can directly convert the amplitude value of the echo signal into the target after compression. In contrast, traditional imaging methods require multiple iterations to predict viable solutions. Data-driven neural network-based methods depend on a vast amount of data to achieve the desired imaging results. In comparison, the proposed deep unfolding imaging network of ASISTA-Net is model driven and uses a convolutional module to learn the physical mapping relationship between the measurement signal and the target scattering coefficient. This approach enables the ASISTA-Net to achieve excellent imaging results with a small training dataset.

**Table 2.** Imaging parameters.

| Methods | MSE | PSNR | SSIM | Run Time |
|---|---|---|---|---|
| ISTA | 0.0035 | 24.08 | 0.65 | 3.14 |
| SBL | 0.0029 | 24.55 | 0.67 | 2.37 |
| VAE | 0.0011 | 29.58 | 0.83 | 0.35 |
| CC-Unet | 0.0009 | 30.45 | 0.85 | 0.27 |
| ASISTA-Net | 0.0006 | 32.21 | 0.91 | 0.10 |

## 5. Discussion

Compared with classical iterative optimization imaging and data-driven end-to-end neural networks, algorithmic unfolding networks can better learn the mapping relationships embodied in the imaging model and boost the imaging error adaptively during the imaging process, thereby ensuring the stability and robustness of such imaging networks. Since the previous simulation was performed in the noiseless case, we trained the proposed network ASISTA-Net with different datasets to study the effects of noiseless and noisy training data on the imaging performance of the network.

The simulation conditions are set to compress the sampling ratio at 0.1 with SNR values of 0 dB, 5 dB, 10 dB, and 20 dB. The noise immunity performance analysis was carried out. In the simulation test, three test scenarios of the same size as the original targets in Figure 5 were selected. The learning rate of the ADAM algorithm was set to $1 \times 10^{-4}$, and the network was trained for 200 epochs.

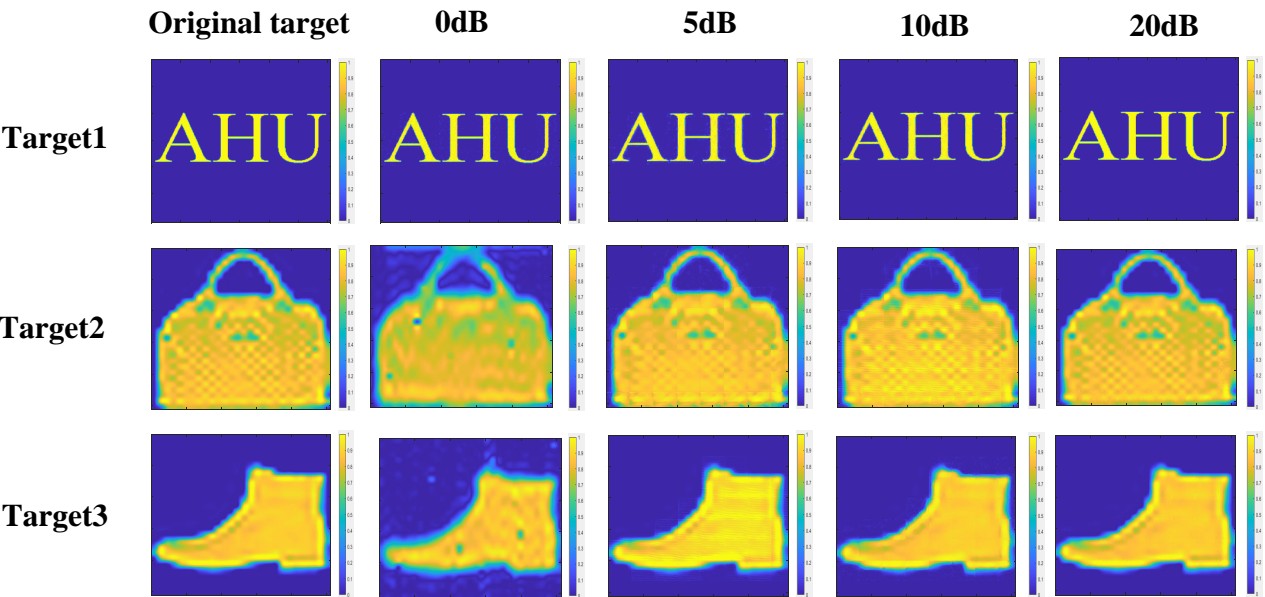

**Figure 5.** Imaging results with different SNR.

To provide more information on network training, we plotted the error curve (loss function) during training iterations to demonstrate the performance of the trained network. Specifically, we showed target 1 in the 20 dB conditions with the training error and validation error as a function of the number of training iterations, which can demonstrate the model convergence, overfitting, and effectiveness of the selected optimization algorithm, among other things. The network loss curve's iterations during the training procedure is depicted. The loss error value exhibits a declining trend and tends to converge quickly, as shown in the figure, further demonstrating the effectiveness of the proposed imaging network.

The qualitative and quantitative analysis of the dataset and imaging reconstructed results under different noise conditions are shown in Figure 6. From the results, the ASISTA-Net can reconstruct targets with high precision and detailed feature information even at a very low SNR of 0 dB. In addition, we recorded and plotted the MSE and PSNR metric values during the imaging reconstruction under these sets of SNR conditions in Figure 7a,b. From the curves, we observed that our imaging network can withstand training data tests regardless of the presence of noise, with insignificant effects on the accuracy of the target reconstruction. In summary, ASISTA-Net is recognized for its excellent noise immunity and robustness.

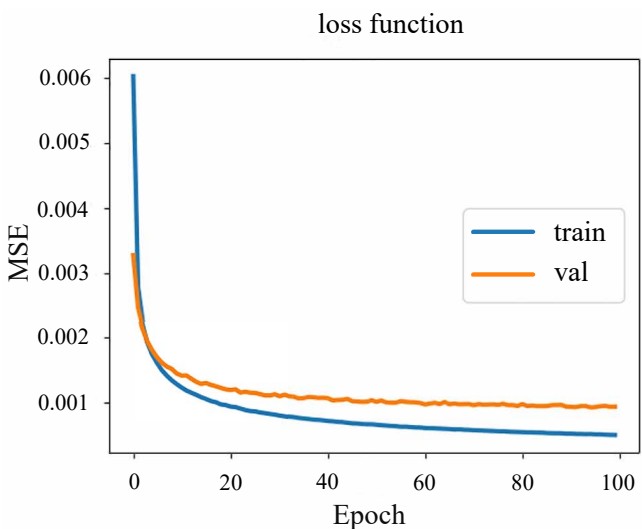

**Figure 6.** Loss function curve of Target 1 at SNR = 20dB.

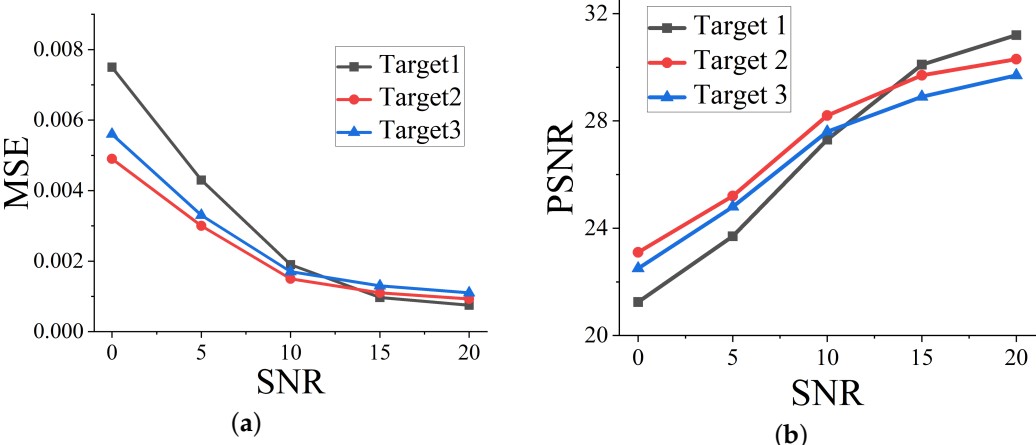

**Figure 7.** Imaging results with different SNR. (**a**) MSE performance. (**b**) PSNR performance.

## 6. Conclusions

This paper introduces ASISTA-Net, a deep unfolding frequency-diverse radar imaging network that incorporates adaptive sampling and an end-to-end model-driven approach. The proposed method shows favorable flexibility and compatibility, enabling reconstruction with the multi-compressed sampling ratio and the reconstruction of multi-scene targets. Through extensive experiments, we demonstrate the robust and effective performance of ASISTA-Net in practical frequency-diverse radar imaging applications. In future research, we intend to improve the quality of our imaging results by further optimizing the model. Additionally, in future research, we focus on evaluating the network's performance using measured datasets and extending the method to imaging 3D scene targets.

**Author Contributions:** Conceptualization, Z.W.; methodology, F.Z.; software, F.Z.; validation, F.Z. and L.Z.; formal analysis, Y.C., J.Q. and J.X.; investigation, Z.W. and F.Z.; resources, L.Y.; data curation, L.Z.; writing—original draft preparation, Z.W.; writing—review and editing, F.Z., L.Z., L.Y., Y.C., J.Q. and J.X.; visualization, Z.W., Y.C.; supervision, L.Y.; project administration, L.Z.; funding acquisition, L.Y. All authors have read and agreed to the published version of the manuscript.

**Funding:** This research was funded by National Natural Science Foundation of China: Grant No.62201007, U21A20457, 62071003, and State Key Laboratory of Complex Electromagnetic Environment Effects on Electronics and Information System (CEMEE) foundation: CEMEE2022Z0302B, and Foundation of An'Hui Educational Committee: No.KJ2020A0026, and Anhui Province University Collaborative Innovation Project: GXXT-2021-028, and Shenzhen Science and Technology Program under Grant: KQTD20190929172704911, and Introduced Innovative Research and Development

Team Project of The Pearl River Talent Recruitment Program under Grant: 2019ZT08X751, and China Postdoctoral Science Foundation: No.2020M681992.

**Data Availability Statement:** The data presented in this study are available on request from the corresponding author.

**Conflicts of Interest:** The authors declare no conflict of interest.

## Abbreviations

The following abbreviations are used in this manuscript:

| | |
|---|---|
| OEWG | Open Waveguide |
| CS | Compressed Sensing |
| SBL | Sparse Bayesian Learning |
| ISTA | Iterative Soft Thresholding Algorithm |
| VAE | Variational Autoencoders |
| MSE | Mean Squared Error |
| PSNR | Peak Signal-to-Noise Ratio |
| SSIM | Structure Similarity Index Measure |
| SNR | Signal-to-Noise Ratio |
| FCNN | Fully Convolutional Neural Network |
| BM3D | Block Matching and 3D Filtering |
| PnP | Plug-and-Play |
| ADAM | Adaptive Momentum Estimation |
| ASISTA-Net | Adaptive Sampling Iterative Soft-Thresholding Network |

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
