# Peer review of "Fast Frequency-Diverse Radar Imaging Based on Adaptive Sampling Iterative Soft-Thresholding Deep Unfolding Network"

_remotesensing, doi:10.3390/rs15133284_

Round 1

Reviewer 1 Report

This paper proposes an innovative adaptive sampling deep unfolding imaging reconstruction network, which achieving faster and more accurate reconstructions than traditional matrix-inversion based iterative algorithms and data-driven deep neural reconstruction networks. Before publishing, I still have the following doubts:

(1) Please provide more information on network training, such as changes in error during training iterations, which will help readers understand the performance of the trained network.

(2) With the increase of SNR, the evaluation indicators have undergone significant changes. Unfortunately, the author only achieved a SNR of 20%. What will happen to a larger SNR.

(3) Please add calculation formulas for MSE, SSIM, and PSNR.

Reviewer 2 Report

In this paper, the authors proposed an adaptive sampling IST deep unfolding network for fast frequency diverse radar imaging. This paper can not be published before a major revision.

1. The novelty of this paper is not well emphasized.

2. What is the relationship between the model in Eq.(8) and the proposed network.?

3. The authors employed the synthesis data in the experiment. Can the authors show some results on the real radar data?

4. How does the proposed method achieve fast frequency diverse imaging?

5. There are many grammatical errors in the paper. Please revise them carefully. 

NA

Reviewer 3 Report

ASISTA-Net, a deep unfolding frequency-diverse radar imaging network that integrates adaptive sampling and an end-to-end model-driven method, is introduced in this interesting research. The proposed method is adaptable and compatible, allowing for multi-compressed sampling ratio reconstruction and multi-scene target reconstruction. The authors demonstrate the robustness and effectiveness of ASISTA-Net in practical frequency-diverse radar imaging applications using substantial experimental results. 

It will be fascinating to see how the technique is expanded to 3D scenarios and how the network's performance is tested using real data sets in future studies.applications. 

The article overall seems well written and worthy of publication. The references are also up-to-date, illustrating the state of the art in research.

Asking for more detail in the description of the methods and presentation of the results, I suggest a re-reading to some inaccuracies: 

- Line 29: Please, replace "scanario" with "scenario".

- Line 31: Please, replace "Speciafically" with "Specifically".

- Line 67: Please, replace "representive" with "representative".

- Line 79: Please, replace "turnig" with "turning".

- Line 89: Please, replace "comparision" with "comparison".

- Line 91: Please, replace "limitted" with "limited".

- Line 92: Please, replace "trainning" with "training".

- Line 94: Please, replace "mearsurement" with "measurement".

- Line 247: Please, replace "Numericial" with "Numerical".

Round 2

Reviewer 1 Report

The author has responded to the reviewer's comments and is willing to accept and publish it